# Unlearning Misalignment for Personalized LLM Adaptation via Instance-Response-Dependent Discrepancies Journal Submissions

**Cheng Chen**                                    *Cheng.chen-16@student.uts.edu.au*
*University of Technology Sydney, Australian Artificial Intelligence Institute*
*FEIT, Australia*
*Center for Frontier AI Research*
*Institute of High Performance Computing*
*Agency for Science, Technology and Research, Singapore*

**Atsushi Nitanda**                                    *atsushi_nitanda@a-star.edu.sg*
*Center for Frontier AI Research*
*Institute of High Performance Computing*
*Agency for Science, Technology and Research, Singapore*
*College of Computing and Data Science, Nanyang Technological University, Singapore*

**Ivor W. Tsang**                                    *ivor_tsang@cfar.a-star.edu.sg*
*Center for Frontier AI Research*
*Institute of High Performance Computing*
*Agency for Science, Technology and Research, Singapore*
*College of Computing and Data Science, Nanyang Technological University, Singapore*

**Reviewed on OpenReview:** *https://openreview.net/forum?id=njE3swFBMc*

## Abstract

While Large Language Models (LLMs) have revolutionized chatbot interactions, they often fall short of aligning responses with the nuanced preferences of individual users, a challenge rooted in the inherently subjective and proprietary nature of those preferences. Consequently, prompt-based learning, though effective in enhancing factual accuracy due to its emphasis on universal correctness, remains insufficient for achieving accurate personalised response alignment. Because user preferences vary widely across individuals and contexts, aligning responses requires a more personalized and context-aware approach. To address this limitation, we propose Consistent Marginalization (CM), a novel framework that aims to unlearn misalignment by constructing a personalised memory bank of instance-response-dependent discrepancies, built from a small set of user preference samples. This personalised memory bank equips LLMs with the ability to understand, recall, and adapt to individual preferences, enabling more consistent and personalized responses. Evaluated across a diverse range of domain-specific datasets and model architectures, CM yields notable improvements in response alignment and robustness. We believe Consistent Marginalization represents a valuable step toward enabling LLMs to become genuinely personable and adaptive conversational agents by understanding user preferences and generating responses that are better aligned with individual user expectations.

## 1 Introduction

Autoregressive large language models (LLMs) have recently achieved widespread adoption due to their remarkable performance across various domains (Brown et al., 2020; Touvron et al., 2023; OpenAI, 2023).

These models are predominantly trained via next-token prediction objectives targeted at maximizing output accuracy (Devlin et al., 2018; Touvron et al., 2023; Hu et al., 2024). However, as LLMs become increasingly employed in personalized applications, requiring alignment with nuanced and individual user preferences, their outputs frequently diverge from anticipated or preferred responses (Zhao et al., 2025; Zhang et al., 2025). Such deviations often characterized as misaligned or undesired outputs, presenting substantial challenges in applications requiring adaptive and personalized supervision, including recommendation systems, personalized response generation, and text mining, where consistency, reliability, and alignment with individual expectations are critical (Wu et al., 2024). While the effectiveness of LLMs is closely tied to the prompt strategies employed , serving as external guidance for self-correction (Brown et al., 2020; Wei et al., 2021; Yao et al., 2022; Liu et al., 2023), these strategies are primarily designed to ensure **output accuracy**, rather than **personalized response alignment**. Figure 1 highlights two separate learning objectives: response alignment (how well an LLM reflects subjective, user-specific preferences) and output accuracy (the objective factual correctness of its content). Standard prompt-based learning (Figure 1a) optimizes for output accuracy but neglects instance-response independent and dependent discrepancies (a misalignment between user intent and model response) illustrated in Figures 1b and 1c. Ignoring these discrepancies leads to persistent misalignment, underscoring the need for mechanisms that can record, recall, and correct personalised response misalignment between the user and LLM. Aligning large language models (LLMs) to generate responses tailored to individual users is challenging because it requires understanding and incorporating user preferences. Unlike complex reasoning tasks, which are objective due to their basis in facts and definitive solutions, user preferences are inherently subjective, reflecting personal opinions and judgments (Hu et al., 2024). While supervised fine-tuning on comprehensive datasets can achieve such alignment, it is often costly, and many user-specific datasets are valuable only to the individual, limiting their applicability to the broader community and countries.

Given a user from a region that uses distinctive slang or localized expressions, it can be challenging for an LLM to accurately interpret their intent, especially when there is no prior knowledge of the user's cultural or linguistic background. Such expressions can easily lead to misinterpretation by the LLM and result in irrelevant or incorrect responses. For example, a user might say, "I want a Kopi O," which in some Southeast Asian regions means to black coffee with sugar. Without cultural grounding, the LLM may misinterpret the query and generate a response that does not match the user's intention. Similarly, local expressions such as "sabo" (a slang term for blame-shifting) or "Can or not?" (meaning "Is this possible?") can confuse the model if it lacks familiarity with these phrases. Without learning the discrepancies, the LLM's responses may become misaligned. To address this, we can collect a small set of representative user preference samples (see Section 4) that include localized expressions or slang (as inputs $X$) along with their intended interpretations (as ground truth outputs $Y$). These user-verified pairs help us estimate the instance–response–dependent discrepancies between the LLM's initial outputs and the user's actual preferences. These discrepancies are stored in a lightweight memory bank (see Section 3.1), which endows the LLM to learn the user's linguistic style and improve future responses. For instance, when the same user later says "Kopi O" or uses expressions like "Let's makan" (meaning "Let's eat"), the LLM can infer to the estimated discrepancy and generate the correct response, such as black coffee or interpreting the phrase as a request to eat, without requiring further clarification. Prompt-based self-reflection and self-correction alone (Zhou et al., 2022; Paul et al., 2023; Shinn et al., 2023; Bang et al., 2023; Li et al., 2023; Zhou et al., 2023) struggle to achieve such personalized alignment, especially when rich user feedback are scarce. Moreover, even in tasks focused on output accuracy, intrinsic self-correction remains limited when models rely exclusively on their own outputs for feedback (Huang et al., 2023; Hu et al., 2024).

Even though existing LLM-based chatbots with long context windows demonstrate some memorization ability, it is often achieved through external, proprietary mechanisms and there is still a lack of persistence across sessions in **API-based settings**, where each query is stateless. In such cases, the context is not retained between sessions, forcing users to repeatedly provide preference information. Furthermore, small or mid-sized companies relying on third-party LLM APIs face significant challenges in offering personalized experiences to users, as they typically lack access to internal memory or user history maintained by the LLM provider. In addition, while some commercial LLMs use prior conversations for personalization, these memory systems are proprietary, non-transparent, and **non-transferable**. Users and developers have no direct access or control over how preferences are stored or updated, nor can they audit or export this

memory to alternative providers or offline deployments. In contrast, our approach maintains a lightweight, user-specific memory of *instance–response–dependent discrepancies*, which can be stored locally or ported between models and platforms. This enables persistent, controllable, and transparent personalization, even in stateless or offline environments. Overall, the goal of this work is to:

- **Equip LLMs to remember and consistently follow each user's preferences, ensuring their responses remain aligned over time.**

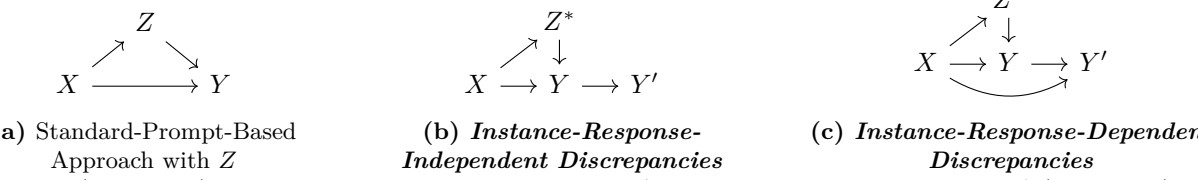

**(a)** Standard-Prompt-Based Approach with $Z$ (equation 1)

**(b)** *Instance-Response-Independent Discrepancies* with Optimal $Z^*$

**(c)** *Instance-Response-Dependent Discrepancies* with Optimal $Z^*$ (equation 3)

Figure 1: **Contrasts standard prompt with Consistent Marginalization (CM) framework and underscores why accounting for instance–response-dependent discrepancies is necessary to bridge the gap between LLM-generated responses and user-preferred responses.** In standard API-based prompt, given an input $X$ to a Large Language Model (LLM), it is assumed that the desired output, or user-preferred responses $Y$, can be obtained by identifying a latent variable $Z$ representing the prompt strategy. Consistent marginalization acknowledges that the LLM's generated responses can deviate from the user-preferred responses $Y$, which is denoted as $Y'$, even for a given input $X$ and an optimal latent variable $Z^*$. Our method explicitly considers these discrepancies and aims to construct a personalised memory bank of the discrepancies between $Y$ and $Y'$, given $X$ and $Z$. We distinguish **two types of response discrepancies**: one of which is **instance-response independent discrepancies** shown in Figure 1(b), where $Y'$ is independent of $X$, and other one is **instance-response dependent discrepancies** shown in Figure 1(c), which is a more realistic scenario. Our study focuses on the latter instance-response *dependent* discrepancies. Assuming that the optimal $Z^*$ has already been determined, our primary learning objective is to estimate $p(Y' \mid Y, X)$. This probability shows the discrepancies between the user-preferred responses $Y$ and the LLM-generated responses $Y'$ under given $X$.

To overcome the limitations, we introduce Consistent Marginalization (CM). This method mitigates misalignment between a large-language-model (LLM) and an individual user's preferred response in a low-data regime without any full user-specific dataset fine-tuning. CM tackles the two fundamental obstacles that prevent off-the-shelf LLMs from delivering truly personalised aligned responses: (i) preference recognition, since current models often fail to detect a user's latent preferences (Zhao et al., 2025), and (ii) preference retention, because even when a preference is conveyed once, the model does not remember it in later turns. We argue that both shortcomings stem from overlooking instance–response–dependent discrepancies during pre-training or the prompt. CM resolves them by (i) explicitly estimating the discrepancy between an LLM's response and the user's preferred response, and (ii) storing these discrepancies in a lightweight "personalised memory bank." During future interactions, the model consults this bank, recalls past misalignments, and self-corrects, achieving strong personalization with only a small set of preference examples and no per-user fine-tuning. Our experimental results validate the effectiveness of the Consistent Marginalization (CM) method. The main contributions include:

- **A general framework for personalised response alignment.** We introduce CM, a method that accounts for instance-dependent discrepancies to mitigate misalignment between an LLM and an individual user's preferred response in data-constrained settings without any full, user-specific fine-tuning.

- **A probabilistic formulation for unlearning misalignment**. CM performs principled, amortised inference of user preferences in a low-data regime by marginalising instance-response discrepancies between the LLM and the user.

- **Demonstrated effectiveness on large-scale datasets with multiple open and closed-sourced LLMS**. We validate our method on multiple user preference-related large-scale datasets. The experimental results demonstrate its effectiveness in enhancing the user personalised response alignment of LLMs without fine-tuning, indicating broad applicability.

## 2 Problem Setting

### 2.1 Preliminaries

Let $X$ denote a user query, expressed as a natural language question. The desired response, provided a priori by the user, is denoted as $Y$. Notably, $Y$ is not generated by the large language model (LLM) but is instead a user-specified reference. A latent variable $Z$ represents the prompt strategy to guide the LLM's generation process. While $Z$ is instrumental in aligning the LLM's response with the user's expectations, it alone is insufficient to guarantee perfect alignment. The LLM-generated response, given $X$ and the chosen $Z$, is represented as $Y'$. Discrepancies between the user desired response $Y$ and the generated response $Y'$ are represented and stored using a deterministic transition matrix $M_{(X)_{Y',Y}}$ (See Section 3.1).

### 2.2 Notation

We define a feature space $\mathcal{X} \subseteq \mathbb{R}^d$ and a label space $\mathcal{Y} = \{1, \ldots, c\}$, where $c$ is the total number of response classes. Each instance $X \in \mathcal{X}$ is associated with a true response $Y \in \mathcal{Y}$ and a generated response $Y' \in \mathcal{Y}$. Here, the user's preferred response $Y$ is treated as the true response. In many real-world scenarios, full supervision is unavailable for the entire dataset. Instead, there is usually a small subset of cleanly labelled samples alongside a larger set of unlabeled data. More specifically, we can define $D_{\text{User}}$ as the distribution of the cleanly labelled small sample set, denoted as user-preference sample, which contains pairs $(X, Y, \vec{Y})$ where $\vec{Y} = \mathcal{Y}$, representing a candidate label set encompassing all responses. This can be expressed as $\{(X_i, Y_i, \vec{Y})\}_{i=1}^s$, with $s$ being the total number of clean samples. We define $D_{\text{large}}$ as the distribution for the large unsupervised dataset, denoted as $\{X_i, \vec{Y}\}_{i=s+1}^n$, where $n$ is the total number of both labelled and unlabeled training samples. The $D_{\text{large}}$ is considered an unsupervised dataset, meaning neither a ground truth response nor weak supervision is associated with each instance in the distribution. The learning objective is to design a prompt strategy that leverages $D_{\text{User}}$, which constitutes about 5% of the total training samples, to allow large language models (LLMs) to accurately annotate the large unsupervised dataset $D_{\text{large}}$. The initial full candidate set $\vec{Y}$ is pruned to a refined candidate set denoted as $C_{\text{refined}} \subseteq \mathcal{Y}$ applying the estimated $M(X)$. The LLM's final response, selected from this refined candidate set, is the **refined prediction**, denoted as $\hat{Y} \in C_{\text{refined}}$.

## 3 *Incorporating* LLM–User Response Dependent Discrepancies via Marginalisation

Most recent work on prompt engineering focus on designing or searching for effective prompt schemes, e.g., chain-of-thought or tool-augmented prompts to maximise the clean (prompt-based) predictive distribution $p(Y \mid X)$ [1] of an LLM (Wei et al., 2022; Yao et al., 2022). The conventional approach of Hu et al. (2024), which *does not* account for personalised response misalignment, is

$$p(Y \mid X) = \sum_Z p(Z, Y \mid X) = \sum_Z p(Y \mid X, Z) \, p(Z \mid X), \tag{1}$$

where $Z$ indexes prompts (latent reasoning paths) and $p(Y \mid X)$ is the clean or prompt-based predictive distribution. Equation equation 1 corresponds to the *ideal*, personalised response misalignment-free setting in which the model's response $Y'$ is always aligned to the user-preferred response $Y$. In practice, however, even an optimal prompt $Z^*$ may be inadequate for personalisation if the LLM has not been pre-trained or fine-tuned on user-specific data. Therefore, the unmodified formulation in equation 1 is inadequate

---

[1] We use "likelihood" informally to denote the model's predictive probability of a sequence (Hu et al., 2024). Where needed, we refer to $p(Y \mid X)$ as the clean/prompt-based predictive distribution and $p(Y' \mid X)$ as the observed-output predictive distribution

for personalised response-alignment tasks because it neglects the systematic discrepancies between actual LLM outputs and user-preferred responses. This work addresses that limitation by explicitly modelling and estimating this discrepancy.

### 3.0.1 Instance–Response Dependent Discrepancies via Marginalisation

We now extend the prompt-based formulation in Eq. equation 1 to a personalised setting where the actual LLM output $Y'$ may be *misaligned* with the user-preferred response $Y$. Let $Y'$ denote the observed (possibly misaligned) response produced by the LLM for input $X$ under prompt $Z$, and let $Y$ denote the latent user-preferred response. By modelling the conditional distribution $p(Y' \mid Y, X)$, we model and later marginalise the *instance–response dependent discrepancy* between $Y'$ and $Y$ given $X$. Under this perspective, the likelihood of observing $Y'$ given $X$ (observed-output predictive distribution) is defined as follow:

$$
\begin{aligned}
p(Y' \mid X) = \sum_Z \sum_Y p(Z, Y, Y' \mid X) &= \sum_Z \sum_Y p(Z \mid X)\, p(Y \mid X, Z)\, p(Y' \mid Y, X, Z) \\
&\stackrel{(i)}{=} \sum_Z \sum_Y p(Z \mid X)\, p(Y \mid X, Z)\, p(Y' \mid Y, X) \\
&= \sum_Y \underbrace{p(Y' \mid Y, X)}_{\substack{\textbf{Instance-Response} \\ \textbf{Dependent Discrepancies}}} \quad p(Y \mid X).
\end{aligned}
\tag{2}
$$

Step (i) assumes the conditional-independence $Y' \perp\!\!\!\perp Z \mid (Y, X)$. This assumption is reasonable, as our goal is to recover an unbiased estimate of LLM misalignment, capturing how the LLM responds to a user query without being undermined by external factors such as the specific prompting strategy. The final factor $p(Y' \mid Y, X)$ is denoted as *instance–response-dependent discrepancies*. Our aim is to estimate the $p(Y' \mid Y, X)$ to maximise the conditional likelihood of LLM in producing personalised responses that align with user preferences. Our formulation explicitly covers the realistic case in which the LLM's raw response $Y'$ is misaligned from the user-preferred response $Y$. When those instance–response-dependent discrepancies are shown, we estimate the $p(Y' \mid Y, X)$ and utilise it to endow the LLM to align more with the user. **In the hypothetical, perfectly aligned regime where $Y' = Y$ for every input, the discrepancy distribution reduces to an identity matrix $p(Y' \mid Y, X) = \mathbf{1}_{\{Y'=Y\}}$ and our framework naturally reduces to equation 1.** Because our learning objective is discrepancy estimation rather than prompt optimisation, we assume a suitable prompt $Z^*$ has already been given. The joint model for the user-preferred response $Y$ and the LLM's misaligned response $Y'$ then factorises as

$$
p(Y' \mid X) = \sum_Y p(Z^*, Y, Y' \mid X) = \underbrace{\sum_Y p(Y' \mid Y, X)}_{\substack{\textbf{Instance-Response} \\ \textbf{Dependent Discrepancies}}} \quad p(Y \mid X)
\tag{3}
$$

To obtain the user's preferred response by searching $Z^*$ is indeed essential but insufficient. In this paper, our goal is to estimate the term $p(Y' \mid Y, X)$, which quantifies the discrepancies between LLM outputs and user preferences for all instances with their corresponding true labels $Y$ by marginalizing over all possible LLM-generated responses $Y'$. Given that $p(Y \mid Z^*, X)$ and $p(Z^* \mid X)$ are either known or given, the task of maximizing $p(Y \mid X)$ ultimately depends on accurately estimating $p(Y' \mid Y, X)$. The $p(Y \mid Z^*, X)$ and $p(Z^* \mid X)$ are assumed obtainable since we assume $Z^*$ is given. $p(Y \mid Z^*, X)$ describes how likely each possible $Y$ is, given the input $X$ and the optimal prompt strategy $Z^*$. However, this distribution alone does not guarantee that the LLM will produce the ground truth response in practice. The model's generated output $Y'$ can still deviate from $Y$ due to inherent uncertainties or imperfections in the LLM. To address this issue, it is crucial to consider the discrepancy distribution $p(Y' \mid Y, X)$ and incorporate it into the inference process. By consistently marginalizing over all possible generated responses $Y'$, we can more accurately model the process by which the true label $Y$ relates to the observed LLM output $Y'$. This leads to a more reliable and coherent inference framework that accounts for the discrepancy between the idealized distribution $p(Y \mid Z^*, X)$ and the practical reality of the LLM's response generation.

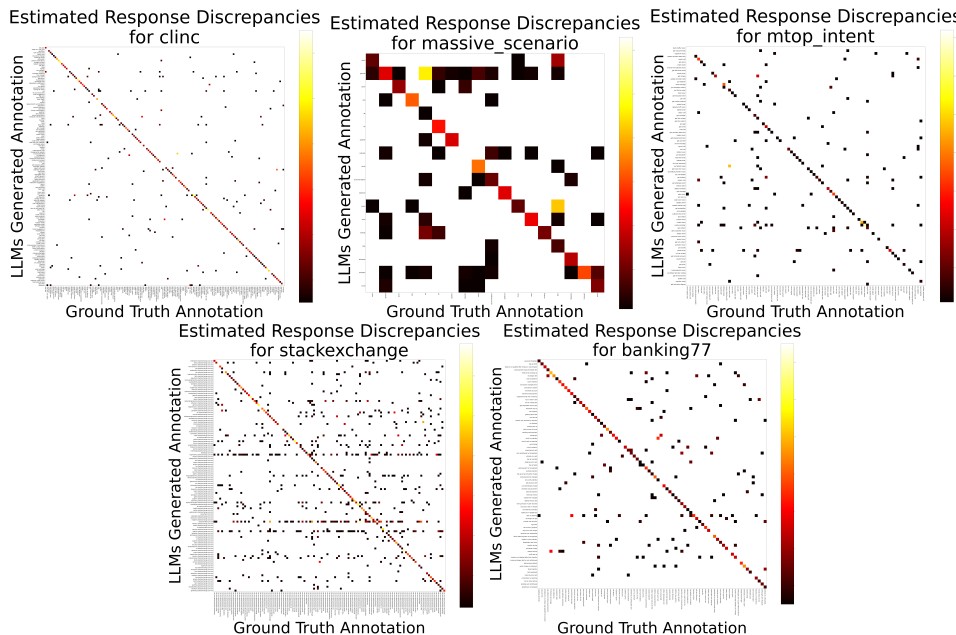

Figure 2: Estimated Instance-Response-Dependent Discrepancies on ChatGPT 3.5 Turbo for datasets StackExchange(Topic) CLINC150(Intent), Banking77(Intent), MOTE(Intent), Massive(Scenario).The diagonal entries of the matrix indicate the aligned responses from LLMs given each dataset, whereas the other highlighted entries indicate misaligned responses. *The annotation also can be denoted as response.*

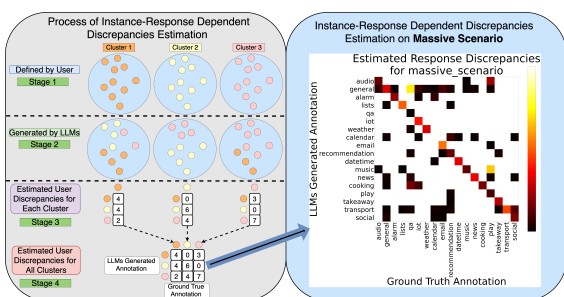

Figure 3: Semantic Depiction of Instance-Response Dependent Discrepancy Estimation. Clusters indicate instances with the same responses.

## 3.1 Instance-Response-Dependent Discrepancies Between LLMs and User

This subsection *bridges theory and implementation*: we show how the notion of response misalignment formalised by $p(Y' \mid Y, X)$ is realised in practice and subsequently exploited.

**Definition of misalignment.** For a fixed query $X$ and ground-truth label $Y$, the probability $p(Y' \mid Y, X)$ which we denote as the *instance–response-dependent discrepancy*—quantifies how often the LLM produces each (possibly mis-aligned) response $Y'$. We instantiate the $p(Y' \mid Y, X)$ via a *deterministic* transition matrix $M(X) \in \{0, 1\}^{|\mathcal{Y}| \times |\mathcal{Y}|}$ whose entry $M(X)_{k',k} = 1$ iff, at least once in the interaction history for query $X$, the LLM produced response $Y' = k'$ while the ground-truth label was $Y = k$. Each column $k$ therefore encodes the set of previously observed alignments ($k' = k$) and misalignments ($k' \neq k$) associated with $Y = k$ given $X$. During our experiments we treat $M(X)$ as an *personalised memory bank* rather than a binary mask. Formally,

$$M(X)_{k',k} = \begin{cases} 1, & \text{if there exists a user preference sample } i \text{ such that } Y_i = k \text{ and } Y_i' = k' \\ 0, & \text{otherwise.} \end{cases}$$

Ideally, only the diagonal entries of $M_{(X)}$ would be positive, indicating that the LLM's generated responses match the true responses $(Y = Y')$. All **off-diagonal** entries would remain zero, indicating perfect response alignment. In practice, however, discrepancies arise which represent instances where the LLM's responses deviate from the true responses in the user preference samples, resulting in **nonzero off-diagonal** elements that capture the discrepancies between true responses and the LLM-generated responses. Each **row** of the estimated matrix $M_{(X)}$ encodes the dependency between the LLM's generated responses $Y'$ and the possible true labels $Y$. Specifically, $M_{(X)}$ records all past discrepancies observed for instances sharing the same true label $Y$. This dependency is utilized to truncate the candidate label set for each sample based on the LLM's responses. Formally, $M$ is defined as a $K \times K$ deterministic transition matrix (where $K$ is the number of classes) with elements $M_{k',k}$, where: For example: $M_{(X)_{1,1}} = 1$ indicates that the LLM predicted responses 1 given the true responses was 1, meaning that there are no response discrepancies. $M_{(X)_{4,1}} = 1$ indicates that the LLM predicted responses 4 given the true responses was 1, meaning that there are response discrepancies. For illustration purposes, consider the following example of the deterministic transition matrix $M_{(X)}$:

$$M_{(X)} = \begin{bmatrix} M_{1,1} & M_{1,2} & M_{1,3} & M_{1,4} \\ M_{2,1} & M_{2,2} & M_{2,3} & M_{2,4} \\ M_{3,1} & M_{3,2} & M_{3,3} & M_{3,4} \\ M_{4,1} & M_{4,2} & M_{4,3} & M_{4,4} \end{bmatrix} = \begin{bmatrix} 1 & 0 & 0 & 1 \\ 0 & 1 & 0 & 0 \\ 0 & 1 & 1 & 0 \\ 1 & 0 & 0 & 1 \end{bmatrix}. \tag{4}$$

The *highlighted entries* denotes previously made misalignments by LLMs recorded for all instances with respect to each true responses $Y$. We treat each true responses as a category, thereby aggregating all possible misalignments of the instances made by LLMs associated with that category. In this deterministic transition matrix, $M_{(X)_{k',k}}$ indicates that the LLM predicted responses $k'$ when the ground-truth responses was $k$ for at least one observed sample. Each **row** of $M_{(X)}$ corresponds to a predicted label $Y' = k'$, and each **column** corresponds to a ground-truth label $Y = k$. The entry $M_{(X)_{k',k}} = 1$ indicates that the LLM has generated the response $Y' = k'$ when the true label was $Y = k$. This can be described as follows:

**Condition 1:** $k' = k$ (Response Alignment). If $k' = k$, the LLM's prediction aligns with the true label, indicating no discrepancy:

$$M_{(X)_{k',k}} = 1 \quad \text{(LLM's prediction matches the true label } Y = k\text{)}.$$

**Condition 2:** $k' \neq k$ (Response Discrepancies). If $k' \neq k$, the LLM has generated an incorrect response $Y' = k'$ for instances where the true label is $Y = k$. This captures a discrepancy:

$$M_{(X)_{k',k}} = 1 \quad \text{(LLM's prediction } Y' = k' \text{ does not match the true label } Y = k\text{)}.$$

By estimating the $M_{(X)}$ from user preference samples $D_{\text{User}}$, we record all possible past misalignments $(Y')$ made by the LLM for instances with the corresponding true response $Y$. This matrix provides a structured representation of the alignment or discrepancies between the LLM's responses and user-preferred responses. Given new inputs, the deterministic transition matrix $M_{(X)}$ and the initial responses from the LLM, we can refine the prediction process as follows: By treating the initial prediction $Y'$ as an index, $M_{(X)}$ allows us to retrieve the set of potential correct responses $Y$ associated with $Y'$ during the estimation phase. This enables the LLM to reconsider its prediction by selecting only from the refined candidate set, effectively recalling past misalignments. To summarise and connect the theoretical construction with our practical implementation, we add the following. The $M(X)_{k',k}$ records every aligned $(k' = k)$ and mis-aligned $(k' \neq k)$ occurrence, we can obtain an *empirical conditional distribution* of misalignment and alignment via simple row normalisation: $p\big(Y' = k' \mid Y = k, X\big) = \widetilde{M}_{(X)k',k} := \frac{M(X)_{k',k}}{\sum_{j \in \mathcal{Y}} M(X)_{j,k}}$ whenever $\sum_j M(X)_{j,k} > 0$. In practice, however, our LLM is a **black-box**: its parameters, loss function, and internal soft-max layer are inaccessible. Subsequently, we cannot train the LLM using estimated $\widetilde{M}_{(X)}$.

**Example: Personalizing LLM Responses to User Preferences** The Figure 4 illustrates overview of CM pipeline and role of instance-response-dependent discrepancy-estimation. Each user query $X_i$ is initially paired with an candidate set $\vec{Y}_i$ that lists *all* potential responses. Our aim is to steer the LLM toward responses that faithfully reflect individual user preferences across the unlabeled corpus $D_{\text{large}}$. **Consistent**

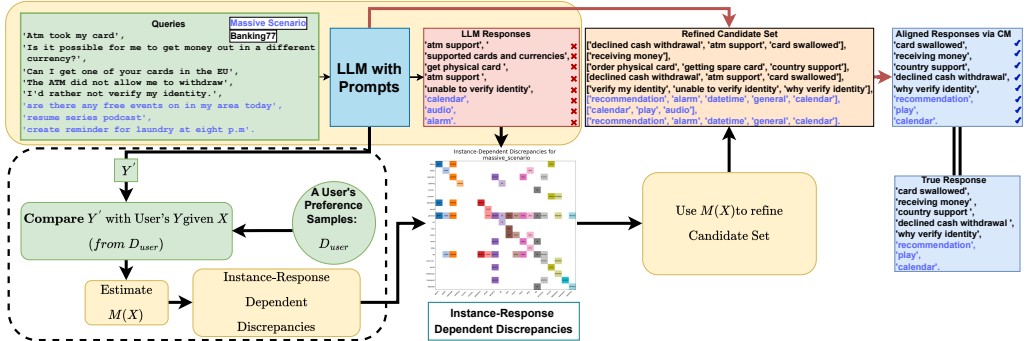

Figure 4: Overview of Consistent Marginalization on Tasks Massive Scenario and Banking77. $D_{\text{User}} = \{(X_i, Y_i)\}_{i=1}^{s}$ denote a small set of user preference samples with size $|s|$. Each input $X_i$ has a known ground-truth response $Y_i$. From these samples, we construct a deterministic transition matrix $M$, which captures mappings between the LLM's generated responses $Y_i'$ and the corresponding ground-truth labels $Y_i$.

**Marginalisation (CM).** CM first *estimates* instance–response-dependent discrepancies between the LLM's misaligned response $Y_i'$ and the ground-truth preference $Y_i$. Because full supervision is impractical, these discrepancies are infers from a small but reliable set of annotated preference samples, storing them in a transition matrix $M_{(X)}$. This matrix acts as a lightweight *personalised memory bank* that records misaligned response of LLM from user's true intent. At inference time, $M_{(X)}$ is exploited to refine the original candidate set $\vec{Y_i}$, generating a pruned set $C_{i_{\text{Refined}}}$ that excludes potentially unwanted responses. Finally, the LLM, represented by the function $G$, generates a refined output $\hat{Y_i}$ by conditioning on the input $X_i$ and the updated candidate set $C_{i_{\text{Refined}}}$ to $\hat{Y_i} = G(X_i, C_{i_{\text{Refined}}})$.

## 4 Selection Criterion for User-Preference Samples

In practice, selecting a small subset of samples (hereafter referred to as "user preference samples") from a large training set is challenging. **High Confidence from Human Annotators:** In parallel, we select samples for which human annotators have high confidence in their preferred responses. Formally, consider an instance $X$ and a label $Y$ drawn from the label set $\vec{Y}$. A user preference sample must satisfy $p(Y \mid X, \vec{Y}) \approx 1$, indicating that the human annotator is certain of their responses. This criterion ensures that the user preference samples are as misalignment-free as possible, minimizing uncertainty and reducing the risk of propagating incorrect user preferences into the learning process. Since our setting assumes the availability of a small set of user-preferred responses, we do not conduct an extensive feasibility study on the selection process. Instead, we adopt this selection criterion as a working assumption to approximate reliability, acknowledging that user-provided preference samples may still contain noise. Nevertheless, we treat these samples as relatively trustworthy compared to LLM generated responses, particularly when they originate from human annotators *Section 4* shows that *instance–response-dependent discrepancies* are essential for aligning LLM response with user preference. We now describe how to estimate $\sum_{Y'} p(Y' \mid Y, X, \vec{Y})$ and specify the assumption of user-preference samples for the estimation under which alignment can be achieved without per-user fine-tuning. Because the candidate set $\vec{Y}$ is deterministic in our setting, it appears explicitly in the conditional notation to ensure the estimation procedure is clear.

**Definition 1** (LLM misaligned Response). Let $G$ denote the LLM. For a query $X$ and candidate set $\vec{Y}$,

$$p(Y' \mid X, \vec{Y}) = G(X, \vec{Y}), \tag{5}$$

where $Y'$ may differ from the user-preferred response $Y$.

**Proposition 1.** Because every model output $Y'$ is paired with (possibly latent) $Y$, we have

$$\underbrace{p(Y' \mid X, \vec{Y})}_{\text{LLM-Generated Response}} = \underbrace{\sum_{Y} p(Y' \mid Y, X, \vec{Y})}_{\substack{\textbf{Instance-Response} \\ \textbf{Dependent Discrepancies}}} \underbrace{p(Y \mid X, \vec{Y})}_{\text{User-Preferred Response}} . \tag{6}$$

**Assumption 1** (High-Confidence User Preference Samples). There exists a finite set

$$\mathcal{D}_{\text{User}} = \{(X_i, Y_i)\}_{i=1}^{s},$$

for which $\Pr(Y = Y_i \mid X_i, \vec{Y}) = 1$ for all $i$.

**Corollary 1** (Discrepancy Estimation). Under Assumption 1, pairing each ground-truth $Y_i$ with its model output $Y_i'$ enables estimation of $p(Y' \mid Y, X, \vec{Y})$.

**Example 1.** If $Y = 1$ is known for $X_1$ with certainty $\left(p(Y = 1 \mid X_1, \vec{Y}) = 1\right)$, then

$$\underbrace{p(Y' = 2 \mid X_1, \vec{Y})}_{\text{LLM-Generated Response (1)}} = \underbrace{\sum_{Y} p(Y' = 2 \mid Y = 1, X_1, \vec{Y})}_{\substack{\textbf{Estimated Instance-Response} \\ \textbf{Dependent Discrepancies (3)}}} \underbrace{p(Y = 1 \mid X_1, \vec{Y})}_{\text{User-Preferred Response (2)}} . \tag{7}$$

With an accurate estimate of the discrepancy distribution $p(Y' \mid Y, X)$, we can learn the clean predictive distribution $p(Y \mid X)$ by matching the $p(Y' \mid X) = \sum_{Y} p(Y' \mid Y, X)\, p(Y \mid X)$, reducing the need for full per-user fine-tuning.

## 5 Experiment

### 5.1 Evaluation Setup

To assess the *effectiveness* and *generalisability* of **Consistent Marginalization (CM)**, we conduct experiments on three widely used LLMs—**Chatgpt-4o-mini**, **Chatgpt-3.5**, and **Llama-8b-Instruct**—each run with two random seeds for robustness. CM is benchmarked against standard prompt and recent self-correction baselines on five diverse, real-world datasets: **StackExchange**: a multi-domain QA corpus (e.g., programming) that tests how well an LLM aligns responses in varied, user-specific contexts. **CLINC150**: 150 intent categories drawn from realistic dialogue, measuring an LLM's ability to capture subtle user preferences in high-variance settings. **BANK77**: banking-themed user queries that probe alignment performance in high-stakes, user-sensitive scenarios. **MOTE**: a multilingual dataset for evaluating CM's cross-lingual adaptability. **Massive Scenario**: 51 typologically-diverse multilingual natural language understanding dataset, highlighting CM's scalability to broad situations and linguistic coverage. For each LLM, we compare every baseline prompt method with its CM-enhanced counterpart under identical conditions that is, *Baseline Prompt* versus CM + *Baseline Prompt* and report results averaged over the two seeds.

### 5.2 Baselines

Self-Consistency Wang et al. (2023) aims to improve the response accuracy of LLMs by considering consistently generated answers through selecting multiple and diverse paths in a few-shot chain of thought approach. The problem with this method is its dependence on multiple sources of paths from the same model; even slight changes in one source's responses can drastically impact the final responses. In addition, self-consistency only reflects the proportions of the dataset used for training the LLMS, but not the personal response preference of the user. The chain of thought method Wei et al. (2022) is a step-by-step illustration for the given query to the LLMs. Few-Shot Thought Prompting: Brown et al. (2020) uses a few relevant examples as illustrations in the prompt to aid the model in aligning the response with the user. We have also included Self-Refine Madaan et al. (2024) to show that relying on model reflection is inadequate to make a model produce persistent alignment.

| Metric
Dataset | ChatGPT-3.5
StackExchange(Topic) | ChatGPT-3.5
CLINC150(Intent) | ChatGPT-3.5
Banking77(Intent) | ChatGPT-3.5
MOTE(Intent) | ChatGPT-3.5
Massive(Scenario) |
|---|---|---|---|---|---|
| CotWei et al. (2022) | $49.82\% \pm 0.19\%$ | $64.62\% \pm 1.63\%$ | $19.16\% \pm 0.14\%$ | $57.42\% \pm 0.40\%$ | $60.54\% \pm 0.24\%$ |
| $\mathbf{CM_{3.5}}$+ **Cot** | $\mathbf{54.72\% \pm 0.17\%}$ | $\mathbf{68.09\% \pm 1.92\%}$ | $\mathbf{31.90\% \pm 0.30\%}$ | $\mathbf{64.91\% \pm 0.36\%}$ | $\mathbf{68.11\% \pm 0.43\%}$ |
| FoTBrown et al. (2020) | $46.04\% \pm 0.01\%$ | $61.62\% \pm 1.16\%$ | $36.77\% \pm 0.44\%$ | $55.93\% \pm 0.23\%$ | $58.23\% \pm 0.45\%$ |
| $\mathbf{CM_{3.5}}$+ **FoT** | $\mathbf{51.92\% \pm 0.08\%}$ | $\mathbf{63.53\% \pm 1.23\%}$ | $\mathbf{45.94\% \pm 0.60\%}$ | $\mathbf{63.94\% \pm 1.15\%}$ | $\mathbf{66.70\% \pm 0.86\%}$ |
| Zero-Shot | $51.96\% \pm 0.02\%$ | $63.18\% \pm 1.13\%$ | $62.55\% \pm 0.85\%$ | $65.84\% \pm 0.47\%$ | $62.22\% \pm 0.05\%$ |
| $\mathbf{CM_{3.5}}$+**Zero-Shot** | $\mathbf{56.29\% \pm 0.04\%}$ | $\mathbf{67.84\% \pm 1.70\%}$ | $\mathbf{66.92\% \pm 1.10\%}$ | $\mathbf{74.36\% \pm 0.73\%}$ | $\mathbf{67.26\% \pm 0.12\%}$ |
| Self-ConsistencyWang et al. (2022) | $51.75\% \pm 0.06\%$ | $68.90\% \pm 0.08\%$ | $56.61\% \pm 0.34\%$ | $68.26\% \pm 0.26\%$ | $62.49\% \pm 0.19\%$ |
| $\mathbf{CM_{3.5}}$+**Consistent** | $\mathbf{53.96\% \pm 0.08\%}$ | $\mathbf{69.36\% \pm 0.19\%}$ | $\mathbf{57.42\% \pm 0.44\%}$ | $\mathbf{75.57\% \pm 0.77\%}$ | $\mathbf{64.63\% \pm 0.12\%}$ |
| Self-RefineMadaan et al. (2024) | $48.94\% \pm 0.68\%$ | $71.63\% \pm 1.24\%$ | $53.90\% \pm 2.94\%$ | $71.88\% \pm 0.59\%$ | $63.55\% \pm 0.02\%$ |
| $\mathbf{CM_{3.5}}$+**Self-Refine** | $\mathbf{54.09\% \pm 0.04\%}$ | $\mathbf{75.29\% \pm 1.23\%}$ | $\mathbf{57.87\% \pm 3.37\%}$ | $68.05\% \pm 0.29\%$ | $\mathbf{68.27\% \pm 0.40\%}$ |

| Metric
Dataset | Llama-8B Instruct
StackExchange(Topic) | Llama-8B Instruct
CLINC150(Intent) | Llama-8B Instruct
Banking77(Intent) | Llama-8B Instruct
MOTE(Intent) | Llama-8B Instruct
Massive(Scenario) |
|---|---|---|---|---|---|
| CotWei et al. (2022) | $14.72\% \pm 0.19\%$ | $32.24\% \pm 0.55\%$ | $22.20\% \pm 0.45\%$ | $55.75\% \pm 0.21\%$ | $46.94\% \pm 0.10\%$ |
| $\mathbf{CM_{Llama}}$+**Cot** | $\mathbf{23.22\% \pm 0.02\%}$ | $\mathbf{33.08\% \pm 0.24\%}$ | $\mathbf{24.62\% \pm 0.19\%}$ | $\mathbf{56.89\% \pm 0.16\%}$ | $\mathbf{50.27\% \pm 0.24\%}$ |
| FoT Brown et al. (2020) | $14.61\% \pm 1.47\%$ | $42.84\% \pm 1.06\%$ | $25.07\% \pm 0.78\%$ | $\mathbf{66.87\% \pm 0.84\%}$ | $54.91\% \pm 1.38\%$ |
| $\mathbf{CM_{Llama}}$+**FoT** | $\mathbf{26.56\% \pm 0.20\%}$ | $\mathbf{43.84\% \pm 0.25\%}$ | $\mathbf{29.00\% \pm 0.35\%}$ | $65.50\% \pm 0.93\%$ | $\mathbf{56.49\% \pm 0.95\%}$ |
| Zero-Shot | $17.80\% \pm 0.19\%$ | $58.24\% \pm 0.69\%$ | $49.03\% \pm 0.13\%$ | $41.09\% \pm 1.42\%$ | $55.82\% \pm 0.19\%$ |
| $\mathbf{CM_{Llama}}$+**Zero-Shot** | $\mathbf{22.66\% \pm 0.09\%}$ | $\mathbf{59.78\% \pm 0.67\%}$ | $\mathbf{55.19\% \pm 0.45\%}$ | $\mathbf{50.91\% \pm 1.77\%}$ | $\mathbf{63.95\% \pm 0.57\%}$ |
| Consistent Wang et al. (2022) | $19.32\% \pm 0.12\%$ | $55.36\% \pm 0.11\%$ | $38.31\% \pm 0.26\%$ | $63.43\% \pm 0.55\%$ | $57.53\% \pm 1.09\%$ |
| $\mathbf{CM_{Llama}}$+**Consistent** | $\mathbf{24.76\% \pm 0.07\%}$ | $\mathbf{55.42\% \pm 0.31\%}$ | $\mathbf{43.34\% \pm 0.35\%}$ | $\mathbf{65.98\% \pm 0.32\%}$ | $\mathbf{59.95\% \pm 0.62\%}$ |
| Self-Refine Madaan et al. (2024) | $18.9\% \pm 0.14\%$ | $60.19\% \pm 0.98$ | $48.70 \pm 0.28\%$ | $42.64\% \pm 1.05\%$ | $55.75\% \pm 0.19\%$ |
| $\mathbf{CM_{Llama}}$+**Self-Refine** | $\mathbf{23.34\% \pm 0.48\%}$ | $\mathbf{61.42\% \pm 1.23\%}$ | $\mathbf{55.23\% \pm 0.32\%}$ | $\mathbf{65.23\% \pm 0.93\%}$ | $\mathbf{63.85\% \pm 0.52\%}$ |

| Metric
Dataset | ChatGPT-4-o-mini
StackExchange(Topic) | ChatGPT-4-o-mini
CLINC150(Intent) | ChatGPT-4-o-mini
Banking77(Intent) | ChatGPT-4-o-mini
MOTE(Intent) | ChatGPT-4-o-mini
Massive(Scenario) |
|---|---|---|---|---|---|
| Cot Wei et al. (2022) | $44.72\% \pm 0.19\%$ | $74.55\% \pm 2.73\%$ | $59.41\% \pm 0.50\%$ | $69.70\% \pm 1.07\%$ | $63.99\% \pm 0.69\%$ |
| $\mathbf{CM_{3.5}}$+**Cot** | $\mathbf{49.54\% \pm 0.31\%}$ | $\mathbf{75.76\% \pm 3.22\%}$ | $\mathbf{65.32\% \pm 1.73\%}$ | $69.25\% \pm 1.32\%$ | $\mathbf{66.40\% \pm 0.38\%}$ |
| FoT Brown et al. (2020) | $42.81\% \pm 0.40\%$ | $73.14\% \pm 2.33\%$ | $55.87\% \pm 0.10\%$ | $66.23\% \pm 0.81\%$ | $67.49\% \pm 0.12\%$ |
| $\mathbf{CM_{3.5}}$+**FoT** | $\mathbf{48.09\% \pm 0.35\%}$ | $\mathbf{74.60\% \pm 2.48\%}$ | $\mathbf{62.74\% \pm 0.89\%}$ | $\mathbf{67.60\% \pm 0.94\%}$ | $\mathbf{70.03\% \pm 0.19\%}$ |
| Zero-Shot | $49.94\% \pm 0.36\%$ | $\mathbf{83.23\% \pm 1.11\%}$ | $66.61\% \pm 1.82\%$ | $\mathbf{73.79\% \pm 0.61\%}$ | $\mathbf{72.04\% \pm 0.07\%}$ |
| $\mathbf{CM_{3.5}}$+**Zero-Shot** | $\mathbf{50.02\% \pm 0.25\%}$ | $82.70\% \pm 1.56\%$ | $\mathbf{69.45\% \pm 2.30\%}$ | $73.72\% \pm 0.94\%$ | $71.03\% \pm 0.07\%$ |
| Consistent Wang et al. (2022) | $47.63\% \pm 0.37\%$ | $\mathbf{81.85\% \pm 0.63\%}$ | $66.08\% \pm 1.38\%$ | $74.00\% \pm 0.32\%$ | $\mathbf{70.82\% \pm 0.02\%}$ |
| $\mathbf{CM_{3.5}}$+**Consistent** | $\mathbf{48.19\% \pm 0.27\%}$ | $79.86\% \pm 1.53\%$ | $65.86\% \pm 1.30\%$ | $\mathbf{79.82\% \pm 0.90\%}$ | $69.81\% \pm 0.40\%$ |
| Self-RefineMadaan et al. (2024) | $\mathbf{51.72\% \pm 0.27\%}$ | $79.34\% \pm 0.49\%$ | $64.81\% \pm 1.33\%$ | $71.93\% \pm 0.02\%$ | $71.35\% \pm 0.29\%$ |
| $\mathbf{CM_{3.5}}$+**Self-Refine** | $51.69\% \pm 0.24\%$ | $\mathbf{80.21\% \pm 1.37\%}$ | $\mathbf{68.71\% \pm 2.68\%}$ | $71.06\% \pm 0.60\%$ | $71.72\% \pm 0.38\%$ |

Table 1: Response–matching accuracy (%±STD) of several prompt methods evaluated on five domain-specific NLP datasets with three LLM back-bones: ChatGPT-3.5-Turbo, ChatGPT-4-o-mini, and Llama-8B-Instruct. For each dataset, the best score obtained by pairing our Consistent Marginalization (CM) module with standard prompt techniques is shown in **bold**. We report Few-Shot, Self-Consistency, Chain-of-Thought(CoT), Self-Refine, and CM variants. *Note:* $CM_{3.5}$ is learned from ChatGPT-3.5-Turbo and then applied to both ChatGPT-3.5-Turbo and ChatGPT-4-o-mini, whereas $CM_{Llama}$ is learned and used solely on Llama-8B. The weaker gains on Chatgpt-4-o-mini highlight that an instance–response discrepancy matrix transfers poorly across model families. In contrast, it delivers substantial improvements when applied to the model on which it was estimated.

| Metric | StackExchange
(Topic) | CLINC150
(Intent) | Banking77
(Intent) | Mtop
(Intent) | Massive
(Scenario) |
|---|---|---|---|---|---|
| Cot Wei et al. (2022) | $14.49\% \pm 0.19\%$ | $32.33\% \pm 0.38\%$ | $22.44\% \pm 0.10\%$ | $55.95\% \pm 0.23\%$ | $47.31\% \pm 0.03\%$ |
| $CM_{Llama}(1\%)$+**Cot** | $\mathbf{19.85\% \pm 0.02\%}$ | $32.18\% \pm 0.04\%$ | $22.14\% \pm 0.06\%$ | $\mathbf{57.32\% \pm 0.14\%}$ | $\mathbf{50.34\% \pm 0.24\%}$ |
| FoT Brown et al. (2020) | $14.73\% \pm 1.54\%$ | $43.02\% \pm 1.38\%$ | $25.45\% \pm 0.45\%$ | $65.50\% \pm 0.57\%$ | $53.30\% \pm 0.03\%$ |
| $CM_{Llama}(1\%)$+**FoT** | $\mathbf{22.52\% \pm 0.38\%}$ | $42.84\% \pm 0.84\%$ | $25.45\% \pm 0.39\%$ | $\mathbf{67.28\% \pm 0.43\%}$ | $\mathbf{56.25\% \pm 0.17\%}$ |
| Zero-Shot | $18.46\% \pm 0.07\%$ | $58.04\% \pm 0.40\%$ | $48.70\% \pm 0.45\%$ | $40.08\% \pm 0.05\%$ | $55.41\% \pm 0.27\%$ |
| $CM_{Llama}(1\%)$+**Zero-Shot** | $\mathbf{22.47\% \pm 0.10\%}$ | $58.22\% \pm 0.22\%$ | $48.38\% \pm 0.52\%$ | $\mathbf{51.12\% \pm 0.23\%}$ | $\mathbf{62.14\% \pm 0.34\%}$ |
| Consistent Wang et al. (2022) | $19.42\% \pm 0.02\%$ | $\mathbf{55.53\% \pm 0.02\%}$ | $38.05\% \pm 0.19\%$ | $62.75\% \pm 0.41\%$ | $56.96\% \pm 0.13\%$ |
| $CM_{Llama}(1\%)$+**Consistent** | $\mathbf{23.75\% \pm 0.31\%}$ | $54.40\% \pm 0.04\%$ | $37.92\% \pm 0.26\%$ | $\mathbf{66.19\% \pm 0.52\%}$ | $\mathbf{59.92\% \pm 0.20\%}$ |
| Self-Refine Madaan et al. (2024) | $18.94\% \pm 0.07\%$ | $58.47\% \pm 0.38\%$ | $48.38\% \pm 0.45\%$ | $41.36\% \pm 0.14\%$ | $55.35\% \pm 0.34\%$ |
| $CM_{Llama}(1\%)$+**Self-Refine** | $\mathbf{22.71\% \pm 0.05\%}$ | $58.27\% \pm 0.22\%$ | $48.21\% \pm 0.55\%$ | $\mathbf{51.03\% \pm 0.05\%}$ | $\mathbf{62.10\% \pm 0.30\%}$ |

Table 2: Response–matching accuracy (%±STD) of several prompt methods evaluated on five domain-specific NLP datasets with Llama-8B-Instruct. For each method (row), the best score across datasets is shown in **bold**. *Note:* $CM_{Llama}(1\%)$ is estimated with 1% of the training sample and used solely on Llama-8B.

## 5.3 Experimental Result

### 5.3.1 Moderate Gains with Just 5 % User-Preference Samples

Consistent Marginalization (CM) is both effective and adaptable. As Table 3 shows, adding CM to every plain-prompt baseline raises response-matching accuracy across all datasets on **Chatgpt-3.5** and **Llama-**

| Model + Method | StackExchange | CLINC150 | Banking77 | MOTE | Massive (Intent) |
|---|---|---|---|---|---|
| **ChatGPT-3.5** | | | | | |
| $CM_{3.5}$ +Cot | +4.90% | +3.47% | +12.74% | +7.49% | +7.57% |
| $CM_{3.5}$ +FoT | +5.88% | +1.91% | +9.17% | +8.01% | +8.47% |
| $CM_{3.5}$ +Zero-Shot | +4.33% | +4.66% | +4.37% | +8.52% | +5.04% |
| $CM_{3.5}$ +Consistent | +2.21% | +0.46% | +0.81% | +7.31% | +2.14% |
| $CM_{3.5}$ +Self-Refine | +5.15% | +3.66% | +3.97% | -3.83% | +4.72% |
| **Llama 8B Instruct** | | | | | |
| $CM_{Llama}$ +Cot | +8.50% | +0.84% | +2.42% | +1.14% | +3.33% |
| $CM_{Llama}$ +FoT | +11.95% | +1.00% | +3.93% | -1.37% | +1.58% |
| $CM_{Llama}$ +Zero-Shot | +4.86% | +1.54% | +6.16% | +9.82% | +8.13% |
| $CM_{Llama}$ +Consistent | +5.44% | +0.06% | +5.03% | +2.55% | +2.42% |
| $CM_{Llama}$ +Self-Refine | +4.44% | +1.23% | +6.53% | +8.77% | +8.10% |
| **ChatGPT-4o-mini** | | | | | |
| $CM_{3.5}$ +Cot | +4.82% | +1.21% | +5.91% | -0.45% | +2.41% |
| $CM_{3.5}$ +FoT | +5.28% | +1.46% | +6.87% | +1.37% | +2.54% |
| $CM_{3.5}$ +Zero-Shot | +0.08% | -0.53% | +2.84% | -0.07% | -1.01% |
| $CM_{3.5}$ +Consistent | +0.56% | -1.99% | -0.22% | +5.82% | -1.01% |
| $CM_{3.5}$ +Self-Refine | -0.03% | +0.87% | +3.90% | -0.87% | +0.37% |

Table 3: Response Alignment Accuracy improvement (%) of Consistent Marginalization (CM) over baseline prompt methods across datasets.

| Method | Objective | Correction Mechanism | Personalised Alignment Ability | Memory of Past Misalignment | External Tool | Limitations on Personalised Response Alignment Task |
|---|---|---|---|---|---|---|
| Few-Shot Brown et al. (2020) | General task performance | None | ✗ Low | ✗ Implicit via reasoning paths | No | Cannot adapt to user-specific preferences or feedback |
| Chain of Thought (Wei et al., 2022) | Improve reasoning accuracy | Implicit via reasoning path | ✗ Low | ✗ None | No | No adaptation to feedback; prone to hallucinations |
| Self-Consistency (Wang et al., 2023) | Improve response accuracy | Majority voting over responses | ✗ Low | ✗ None | No | Relies on diverse paths; No user-specific guidance |
| Self-Refine (Madaan et al., 2024) | Iterative self-correction with Feedback | Self-generated revision loop | Medium | ✗ None | Yes | Requires Many Iterative rounds of Correction. Lack of long-term Memory |
| **Consistent Marginalization (Ours)** | Personalized response alignment | Explicit memory of instance-response dependent discrepancies | ✓High | ✓ Yes | No | **No need for per-user Fine-tune; scalable and adaptive to personalised user preferences** |

Table 4: Comparison between Prompt-Based Methods and Consistent Marginalization (CM) on Personalized Response Alignment Task

**8b-Instruct**, and delivers notable gains on **Chatgpt-4o-mini**. **ChatGPT-3.5-Turbo.** CM improves every prompt strategy on four of the five datasets; the single dip occurs with CM+Self-Refine on MOTE. The largest boost is **+12.74 %** on BANKING77 with CM+CoT, while the other datasets record balanced gains of +3–8 %. **Llama-8B-Instruct.** CM produces the strongest overall gains: 23 of 25 cases improve, 16 by at least 2 %. The highest is **+11.95 %** on STACKEXCHANGE with CM+FoT; the single drop is a mild −1.37 % on MOTE with CM +FoT. **ChatGPT-4o-mini.** Because the discrepancy matrix was learned on GPT-3.5-Turbo, gains transfer slightly less, yet CM still adds **+6.87 %** on BANKING 77 with CM+FoT and remains positive or neutral on most other cells. Overall, with just 5% of user-preference samples, CM consistently outperforms every plain-prompt baseline, confirming its practical value across prompts, datasets, and LLM architectures.

### 5.3.2 Robust Gains with Just 1% User-Preference Samples

To quantify how many annotated preferences CM needs, we perform an ablation using **Llama-8b-Instruct** and two annotation budgets **1**% and **5**% of the training set across all five datasets (Table 2). **Robust gains with only 1 %.** Even at the 1 % budget, CM lifts nearly every prompt baseline. The largest improvement occurs on STACKEXCHANGE, where CM raises *FoT* from 14.73 % to 22.52 % (+7.79 pp) and *Consistent* from 19.42 % to 23.75 % (+4.33 pp). These results show that CM is effective in low-data settings. Overall, CM scales smoothly with additional data while remaining highly effective in resource-constrained regimes.

**Additional LLM: ChatGPT-4 Turbo.** To test the cross-model consistency of CM, we transfer the instance–response discrepancies estimated on ChatGPT-3.5-Turbo to **ChatGPT-4 Turbo**. As Table 5 shows, this transfer still lifts the *Zero-Shot Prompt* baseline by **3.5%** in response alignment ratio, showing that $CM_{3.5}$ remains useful even on a stronger backbone. However, the absolute gain is smaller than on ChatGPT-3.5-Turbo. We attribute the drop to model architecture and parameter difference : the larger the gap between the source model (where the discrepancies were learned) and the target model, the

| Method | Ratio (%) |
|---|---|
| Zero-Shot (GPT-3.5) | 62.99 |
| **$CM_{3.5}$+Zero-Shot** | **67.35** |
| Zero-Shot (GPT-4 Turbo) | 61.98 |
| **$CM_{3.5}$+Zero-Shot** | **64.48** |

Table 5: Cross-LLMs Consistency of CM

less precisely those discrepancies characterise the new model's misalignment.

This observation indicates that the effectiveness of a instance response dependent discrepancy decays as the underlying LLM family shift farther from the one on which it was estimated.

# 6 Related Works

## 6.1 Prompt-based learning

Prompt-based learning, first popularised by Brown et al. (2020), shows that a handful of in-context examples can nudge an LLM toward more accurate responses. Subsequent prompt-engineering methods—instruction tuning (Wei et al., 2021), chain-of-thought (Cot) prompt Wei et al. (2022), and ReAct Yao et al. (2022)—inject explicit task structure or external tool calls to improve reasoning further. Active Prompt Diao et al. (2023), Generate-Knowledge Prompting Liu et al. (2023), and Consistency Prompting Wang et al. (2023) curate clarifying questions, external facts, or multi-path consensus so that the final answer is more consistent and accurate. A parallel line of research explores *self-correction*: Self-Refine Madaan et al. (2024) and Tree-of-Thought (Tot) Long (2023); Yao et al. (2024) allow the LLM to iteratively critique and revise its own outputs. Reflexion Shinn et al. (2023) introduces external tools to validate self-generated feedback and maximise accuracy, but obtaining tools that can assess personalised response alignment at scale remains challenging. Moreover, Huang et al. (2023) shows that self-generated feedback can actually reduce quality when no external check is available, making previous methods less effective for personalised response alignment. However, these methods still fall short on two fronts that are central to personalised response alignment: (i) they do not tackle fine-grained *preference recognition* at the instance level, and (ii) they lack a *memorisation mechanism* that would let the LLM remember and reuse past preference signals over time. Our work addresses both gaps.

## 6.2 User Instance-Response-Dependent Discrepancies Estimation

Patrini et al. (2017); Han et al. (2018); Yang et al. (2022) leverage instance-*independent and dependent* label transition matrices to achieve consistent classifiers. These transition matrices, which resemble response instance-independent and dependent discrepancies. However, such methods generally rely on a **white-box setting**, where model parameters are directly accessible. This accessibility enables the estimation of transition matrices in *probabilistic forms* rather than discrete forms. Despite their effectiveness in white-box scenarios, these approaches have not addressed the accurate estimation of instance-response dependent discrepancies, which capture finer-grained relationships between model outputs and user preferences. Addressing this gap is essential for enabling robust, personalized response alignment in real-world applications, particularly when working with sourced **black-box models** such as open and close-sourced large language models (LLMs).

# 7 Conclusion

We have introduced Consistent Marginalization (CM), a paradigm that enables large language models to deliver personalised responses by explicitly modelling instance–response–dependent discrepancies. CM's learning objective stores each misalignment between the model's draft and the user-preferred answer, then recalls this memory to "unlearn" past errors and adapt future outputs. With only a small set of annotated preference examples and no per-user fine-tuning, CM identify and corrects these discrepancies, steadily steering the LLM toward the user's desired response. Experiments across multiple LLM backbones and five diverse, large-scale datasets show consistent gains in response alignment, confirming CM's practicality and robustness in data-constrained settings.

**Broader Impact Statement**

To the best of our knowledge, our research does not introduce any negative societal or ethical consequences.

**Acknowledgments**

This research is supported by the National Research Foundation, Singapore under its National Large Language Models Funding Initiative (AISG Award No: AISG-NMLP-2024-003). Any opinions, findings and conclusions or recommendations expressed in this material are those of the author(s) and do not reflect the views of National Research Foundation, Singapore.

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

# 8 Appendix

**Table of Notations**

Table 6: Summary of notations. We use lowercase $p(\cdot)$ for (conditional) probability distributions.

| Symbol | Description |
|---|---|
| $X$ | Input instance / user query. |
| $Z$ | Prompt / latent reasoning variable. |
| $Z^*$ | A fixed (e.g., optimal) prompt; if used, $p(Z^* \mid X) = 1$. |
| $Y$ | User-preferred (clean, latent) response / class. |
| $Y'$ | Observed LLM output (possibly misaligned). |
| $\hat{Y}$ | Final aligned response after correction/refinement. |
| $\vec{Y}$ | Initial candidate set (often the full label set $\mathcal{Y}$). |
| $C_{\text{refined}}$ | Refined candidate set pruned from $\vec{Y}$. |
| $\mathcal{Y}$ | Label space (set of classes). |
| $C$ | Number of classes ($C = |\mathcal{Y}|$). |
| $G$ | Large language model (generator). |
| $p(Y \mid X, Z)$ | Clean (prompt-based) predictive distribution under prompt $Z$. |
| $p(Y \mid X)$ | Prompt mixture: $\sum_Z p(Z \mid X)\, p(Y \mid X, Z)$ (or $p(Y \mid X, Z^*)$ if $Z^*$ fixed). |
| $p(Y' \mid Y, X)$ | Instance–response discrepancy (misalignment) distribution. |
| $p(Y' \mid X)$ | Observed-output likelihood: $\sum_Y p(Y' \mid Y, X)\, p(Y \mid X)$. |
| $D_{\text{User}}$ | Labeled user-preference dataset (clean labels). |
| $D_{\text{Large}}$ | Large-scale unlabeled (or weakly labeled) dataset. |
| $s$ | Number of labeled samples ($|D_{\text{User}}|$). |
| $n$ | Total training samples ($|D_{\text{User}}| + |D_{\text{Large}}|$). |
| $Y' \perp Z \mid (Y, X)$ | Independence assumption used in derivations (discrepancies don't depend on prompt given $(Y, X)$). |

## 8.1 Example:Instance-Response Dependent Discrepancy Estimation

Let $D_{\text{User}} = \{(X_i, Y_i)\}_{i=1}^{s}$ denote a small set of user preference samples, where $s = 4$. Each input $X_i$ has a known ground-truth label $Y_i$. From these samples, we construct a deterministic transition matrix $M$, which captures mappings between the LLM's generated responses $Y_i'$ and the corresponding ground-truth labels $Y_i$. We also define the unlabeled dataset as $D_{\text{large}} = \{X_i\}_{i=1}^{N}$, with $N = s = 4$ in this illustrative example. Let $\vec{X} = \{X_1, X_2, X_3, X_4\}$ be the set of input queries, and let $\vec{Y}_{\text{Query}} = \{\vec{Y}_1, \vec{Y}_2, \vec{Y}_3, \vec{Y}_4\}$ denote the initial candidate sets for each query. Assuming a known label space of size $K$, each $\vec{Y}_i$ is initialized to the same universal candidate set $\vec{Y} \in \{0, 1\}^K$, where all entries are initially set to one—i.e., every class is a potential candidate. We define $Y_{\text{True}} = \{Y_1, Y_2, Y_3, Y_4\}$ as the set of one-hot encoded vectors representing ground-truth responses. Using this setup, the personalised memory bank $M(X)$ is estimated from $D_{\text{User}}$ and subsequently used to refine the prediction space for the LLM during inference, producing a final output $\hat{Y}_i = G(X_i, C_{i_{\text{Refined}}})$.

$$
\vec{Y}_{\text{Query}} = \begin{bmatrix} 1 & 1 & 1 & 1 \\ 1 & 1 & 1 & 1 \\ 1 & 1 & 1 & 1 \\ 1 & 1 & 1 & 1 \end{bmatrix} \quad Y' = \begin{bmatrix} 0 & 0 & 0 & 1 \\ 0 & 1 & 0 & 0 \\ 0 & 0 & 0 & 1 \\ 1 & 0 & 0 & 0 \end{bmatrix} \quad \vec{Y}_{\text{G}} = \begin{bmatrix} 1 & 0 & 0 & 0 \\ 1 & 0 & 0 & 0 \\ 0 & 0 & 1 & 0 \\ 0 & 0 & 0 & 1 \end{bmatrix} \quad M_{(X)} = \begin{bmatrix} 1 & 0 & 0 & 1 \\ 1 & 1 & 0 & 0 \\ 1 & 0 & 1 & 1 \\ 1 & 0 & 0 & 1 \end{bmatrix}
$$

$$
\begin{bmatrix}
\text{Given LLMs Generated Response } Y_i' \text{ on } \vec{X}_i \\
\text{If LLM generated response is } \boxed{1} \qquad \begin{bmatrix} 1 & 0 & 0 & 0 \\ 1 & 0 & 0 & 0 \\ 0 & 0 & 1 & 0 \\ 0 & 0 & 0 & 1 \end{bmatrix} \\
\text{If LLM generated response is } \boxed{2} \\
\text{If LLM generated response is } \boxed{3} \\
\text{If LLM generated response is } \boxed{4}
\end{bmatrix}
$$

In the right-hand-side matrix, the estimated $M_{(X)}$ represents the potential user-preferred response $\vec{Y}_{i_{\text{Updated}}}$ for $\vec{X}_i$ corresponding to different LLM-generated responses. $\vec{Y}_{\text{G}}$ denotes the ground truth responses with respect to $X$. If the LLM generates response $Y' = k$, then $\vec{Y}_{i_{\text{Updated}}}$ will be the $k$-th row of the matrix $M_{(X)}^{Y_1}$.

This rule applies to all LLM-generated responses and significantly reduces the candidate set of responses from $[1, 1, 1, 1]$ to a one-hot vector (e.g., $[1, 0, 0, 0]$ if $k = 1$).

**Inference**

During inference, the goal is to leverage the learned discrepancy matrix $M(X)$ to guide the LLM's predictions on the unlabeled dataset $D_{\text{Large}}$. For each unannotated instance $X_i$, the initial candidate set $\vec{Y}$ is pruned using the model's learned error patterns to create a **refined candidate set**, $C_{i,\text{refined}}$. The LLM, denoted as $G$, then takes the instance $X_i$ and this constrained set $C_{i,\text{refined}}$ as input to produce the final, **refined prediction** $\hat{Y}_i$:

$$\hat{Y}_i = G(X_i, C_{i\text{Refined}})$$

The objective is to ensure that this final prediction $\hat{Y}_i$ accurately matches the true response $Y_i$.

## 8.2 Comprehensive comparison table showing the performance differences between 1% and 5% user-preference samples across all methods and datasets

| Dataset | Method | 1% Acc | 5% Acc | Improvements |
|---|---|---|---|---|
| StackExchange | Cot | 14.49 | 14.72 | +0.23 |
| | CM+Cot | 19.85 | 23.22 | **+3.37** |
| | FoT | 14.73 | 14.61 | -0.12 |
| | CM+FoT | 22.52 | 26.56 | **+4.04** |
| | Consistent | 19.42 | 19.32 | -0.10 |
| | CM+Consistent | 23.75 | 24.76 | **+1.01** |
| CLINC150 | Cot | 32.33 | 32.24 | -0.09 |
| | CM+Cot | 32.18 | 33.08 | +0.90 |
| | Zero-Shot | 58.04 | 58.24 | +0.20 |
| | CM+Zero-Shot | 58.22 | 59.78 | **+1.56** |
| | Self-Refine | 58.47 | 60.19 | **+1.72** |
| | CM+Self-Refine | 58.27 | 61.42 | **+3.15** |
| Banking77 | Cot | 22.44 | 22.20 | -0.24 |
| | CM+Cot | 22.14 | 24.62 | **+2.48** |
| | FoT | 25.45 | 25.07 | -0.38 |
| | CM+FoT | 25.45 | 29.00 | **+3.55** |
| | Zero-Shot | 48.70 | 49.03 | +0.33 |
| | CM+Zero-Shot | 48.38 | 55.19 | **+6.81** |
| | CM+Self-Refine | 48.21 | 55.23 | **+7.02** |
| Mtop | Cot | 55.95 | 55.75 | -0.20 |
| | CM+Cot | 57.32 | 56.89 | -0.43 |
| | FoT | 65.50 | 66.87 | **+1.37** |
| | CM+FoT | 67.28 | 65.50 | -1.78 |
| | Consistent | 62.75 | 63.43 | +0.68 |
| | CM+Consistent | 66.19 | 65.98 | -0.21 |
| Massive | Cot | 47.31 | 46.94 | -0.37 |
| | CM+Cot | 50.34 | 50.27 | -0.07 |
| | FoT | 53.30 | 54.91 | **+1.61** |
| | CM+FoT | 56.25 | 56.49 | +0.24 |
| | Zero-Shot | 55.41 | 55.82 | +0.41 |
| | CM+Zero-Shot | 62.14 | 63.95 | **+1.81** |
| | CM+Self-Refine | 62.10 | 63.85 | **+1.75** |

Table 7: Accuracy comparison between 1% and 5% user-preference samples. Bold indicates improvements >1%.

# 9 How Does Consistent Marginalisation Handle Out-of-Domain Questions

While inputs are inherently free-form, in our experimental setup, the output space is predefined and discrete, where responses are mapped to specific categories (e.g., banking intents, user queries in StackExchange). We acknowledge, however, that in real-world settings, users may issue queries from new domains and exhibit previously unseen preferences. There are two possible scenarios for such new input cases:

- (1) new inputs associated with an existing predefined class,

- (2) new inputs associated with a new, previously unseen class.

(1): In our experiments, instance–response–dependent discrepancies are estimated based on user preference samples, which are drawn independently and identically distributed (i.i.d.) from the training distribution. We have verified that our method performs effectively on new inputs (from the testing distribution) that fall within the same set of predefined classes.

(2): Nonetheless, if the new input comes from an out-of-domain distribution which is drastically different from the training data, we can adapt CM to this case with the help of a lightweight pretrained model. To support progressive personalization, our method can be designed to be continually updatable: When a new input $X_{\text{new}}$ is given, which can be significantly different from our in-domain datasets, we can use a pretrained sentence encoder to compute the similarity between the embedding of $X_{\text{new}}$ and those of existing user-preference samples with predefined candidate sets. If the similarity score is lower than a predefined threshold (e.g., 0.1), the system flags $X_{\text{new}}$ as a potential out-of-domain sample. The system then passes the query to the LLM without showing predefined classes. If user feedback is provided either through explicit correction or satisfaction signals the user may reject the undesired response $Y'$ and supply a preferred alternative $Y$. This new instance–response pair is incorporated into the memory $M(X)$ without retraining the LLM. If the user is satisfied, the memory bank records the preference as aligned; otherwise, it logs the misalignment. This setup supports continuous learning without fine-tuning and enables enhanced personalization via a lightweight, per-user memory.

