# OpenReview forum: "Unlearning Misalignment for Personalized LLM Adaptation via Instance-Response-Dependent Discrepancies"
_TMLR — Accepted by TMLR_

### Review · Reviewer_KmLT · 2025-06-14

**Summary Of Contributions:**

The goal of the paper is to resolve user preference and LLM output misalignment issues. To do this, the paper proposes estimating a consistent marginalization term by using a memory bank describing the user preferences. The personalized memory bank is a binary matrix where the (i, j)-th element is 1 if the i-th misaligned response leads to j-th user-aligned response. During inference, the matrix can be utilized as a state transition matrix to reach to the aligned response from the unconditioned LLM response. Experiments have been performed on several LLMs and tested on 5 human preference alignment tasks. In the tables, the results show positive improvements in most cases except the ChatGPT-40-mini experiments where the re-aligned response perform at about same accuracy level as the unaligned responses (better baseline, improvements remain limited).

**Audience:**

Yes

**Broader Impact Concerns:**

Probably no specific problem is introduced by the model itself other than the general issues with LLMs.

**Claims And Evidence:**

Yes

**Requested Changes:**

Please review the formulation of the equations. Do we really need to estimate $ \sum_{Y'} Y' p(Y' | Y,  X)$?

**Strengths And Weaknesses:**

Strengths:

1. Mostly positive results

2. Approach shows some amount of generalizability to other LLMs

3. A practical approach

Weaknesses:

1. One main point that I could not understand is why do we need to estimate $ \sum_{Y'} Y' p(Y' | Y,  X)$? Isn't it always 1 by definition? Is there a notation issue here that I am missing? This appears in Eq. (4).

2. If there is no need to estimate  $ \sum_{Y'} Y' p(Y' | Y,  X)$ why the paper mentions this component? I see that what is actually estimated is the memory matrix $M$ which is simply done by counting and normalizing the co-occurences of response $i$ and $j$

3. From the matrix definition of the memory, I think that the alignment correction can be only useful if the test set covers similar types of questions as the training set. If that is the case, this is a major limitation where we cannot ask an out-of-domain question and add a new preference and expect alignment correction.

---

> ### Author Response · Authors · 2025-07-19
> ****Response to Reviewer KmLT****
>
> Dear Reviewer,
>
> Thank you for your thoughtful and detailed feedback. We address your main concern and follow-up clarification below.
>
> ---
>
> #### **W1: One main point that I could not understand is why do we need to estimate $\sum_{Y'}Y' p(Y' \mid Y, X)$? Isn't it always 1 by definition? Is there a notation issue here that I am missing? This appears in Eq. (4).**
> Thank you for the comment. We believe there may have been a misunderstanding in the interpretation of our formulation. The expression you cited, $\sum_{Y'} Y' p(Y' \mid Y, X)$, does **not** appear in our paper. Our goal is **not** to estimate this expected value. Instead, we emphasize the importance of estimating the conditional distribution $p(Y' \mid Y, X)$, which captures the **instance–response–dependent discrepancy** between the LLM-generated response and the user's preferred response. This modeling of discrepancy is essential and is typically omitted in conventional prompt-based approaches.
>
> ---
>
> #### **W2: Why does the paper mention $\sum_{Y'} Y' p(Y' \mid Y, X)$?**
> We believe there may have been a misunderstanding in the interpretation of our formulation. Our goal is **not** to estimate the quantity $\sum_{Y'} Y' \, p(Y' \mid Y, X),$ which would imply computing an expected value over all possible responses. Instead, our objective is to emphasize the estimation of the **instance–response–dependent discrepancy**, specifically $p(Y' \mid Y, X),$ which captures how a LLM response $Y'$ diverges from a user-preferred response $Y$ given the input instance $X$. This instance -response dependent discrepancy forms the foundation of our Consistent Marginalization (CM) framework.
>
> ---
>
> #### **W2.1: Clarification on memory matrix $M$ and how it relates to $p(Y' \mid Y, X)$**
> The relationship between the probabilistic term and our memory matrix is straightforward: the probability distribution $p(Y' \mid Y, X)$ is the **theoretical justification** for CM, whereas the memory matrix $M(X)$ is its **practical implementation**. Since we treat the LLMs as **black-box** models that produce **discrete responses**, we can't access their internal likelihoods directly. This constraint makes it essential to estimate their predictive behavior empirically. The memory matrix $M(X)$ is the mechanism we designed for this purpose. It works by capturing the alignment and misalignment between the LLM's predicted response $k'$ and the true response $k$. Normalizing the co-occurrence counts ensures our method is theoretically complete.
>
> ---
> #### **W3: From the matrix definition of the memory, I think that the alignment correction can be only useful if the test set covers similar types of questions as the training set. If that is the case, this is a major limitation where we cannot ask an out-of-domain question and add a new preference and expect alignment correction.**
>
> If the new input comes from an out-of-domain distribution that is drastically different from the training data, we can adapt CM to this case with the help of a lightweight pretrained model. To support **progressive personalization**, our method can be designed to be **continually updatable**.
> *When a new input $X_{\text{new}}$ is given, which may significantly differ from our in-domain datasets, we use a pretrained sentence encoder to compute the similarity between the embedding of $X_{\text{new}}$ and those of existing user-preference samples with predefined candidate sets. If the similarity score is lower than a predefined threshold (e.g., 0.1), the system flags $X_{\text{new}}$ as a potential out-of-domain sample.*
> The system then passes the query to the LLM without showing predefined classes. If user feedback is provided, either through explicit correction or satisfaction signals, the user may reject the undesired response $Y'$ and supply a preferred alternative $Y$. This new instance–response pair is incorporated into the memory $M(X)$. If the user is satisfied, the memory bank records the preference as aligned; otherwise, it logs the misalignment. This setup supports **continuous learning without fine-tuning** and enables enhanced personalization through a lightweight, **per-user memory**. We will detail this generalization strategy and its limitations in the revised manuscript and acknowledge that extending this alignment to truly **OOD scenarios is a valuable direction for future work**. However, we considered it to be **outside the immediate scope** of this paper
>
> ---
> #### **Requested Changes:**
> Please refer to the response for W1 and W2.

---

### Review · Reviewer_DHBS · 2025-06-17

**Summary Of Contributions:**

This paper studies the problem of personalization in LLM. The key idea is to model the differences between model output and expected output as a probabilistic model, which contains the user input X, user truly desired output Y, the hidden context Z that affects X, Y and the model's output Y'. The paper argues that previous work usually considers dependency between Y' and Y, where this work also added dependency between X and Y'. To model the probabilistic distribution, it employs an adaption matrix that encodes user preference from past logs with golden user choice. Named as Consistent Marginalization (CM), the algorithm can be appled on top of other personalization methods.

The paper evaluates the performance of CM with a few popular models (gpt3.5, gpt4o-mini and Llama-8b-instruct) on several datasets (StackExchange, CLINC150, BANK77, MOTE and Massive Scenario). Evaluation shows that when added with CM, most existing algorithm has a better performance except in a few cases. The paper further performs ablation to understand the effects of different amounts of golden user preference used, claiming that even with a smaller amount (1% vs 5%) it still retains effectiveness.

**Audience:**

Yes

**Broader Impact Concerns:**

No concern.

**Claims And Evidence:**

Yes

**Requested Changes:**

See weakness part.

**Strengths And Weaknesses:**

The strength of the paper is mostly in extensive experiments on a wider array of models, datasets and baseline algorithms.

The weakness are as follows, ordered by reviewer's level of concern:

1) It's not clear how the matrix M can generalize to new input. All description assumes that X, Y and Y' are categorical in a finite set, where LLM input and output are freeform and can't be directly compared. The reviewer may have missed something but an average reader would too.

2) The claim that LLM is ineffective in personalization is not convincing. With in-context learning (where you can put user preference as part of the prompt) and fine-tuning (esp. LoRA), it's widely accepted that LLMs can be personalized or contextualized with reasonable compute.

3) The paper has large chunks of seeming redundant content, that can be safely trimmed off. For example most of page 6 and 7.

Other less concerned weakness that could be easily fixed.

1) In paragraph 1, the definition of "hallucination" is usually not the misalignment between desired and actual output.

2) The paper should explain what is X, Y, Z at the very beginning with some concrete example, at least move 3.1 to the very beginning, or it's hard to get a concrete idea.

3) Is Y at token level or response level?

4) Page 8 second line of equation, what is this?

5) Page 9 at the top: it's not practical to assume reliability from user-provided preference. There is always noise.

6) Better annotate the experiment results with statistical significance (bold only the significant differences) and which one is state-of-the-art.

7) How is CI computed?

---

> ### Author Response · Authors · 2025-07-20
> **Response to Reviewer DHBS**
>
> Dear Reviewer,
>
> Thank you very much for your detailed and constructive feedback.
>
> ---
>
> #### **W1 It's not clear ,..., directly compared.**
>
> While inputs \$X\$ are inherently freeform, in our experimental setup, **the output space \$Y\$ is predefined and discrete**, where responses are mapped to specific categories (e.g., banking intents, user queries in StackExchange). We acknowledge, however, that in real-world settings, users may issue queries \$X\$ from new domains and exhibit previously unseen preferences.
>
> There are two possible scenarios for such new input cases:
>
> (1) *new inputs associated with an existing predefined class*,
>
> (2) *new inputs associated with a new, previously unseen class*.
>
> (1): In our experiments, instance–response–dependent discrepancies are estimated based on user preference samples, which are drawn independently and identically distributed (i.i.d.) from the training distribution. We have verified that our method performs effectively on *new inputs* (from the testing distribution) that fall within the same set of predefined classes.
>
> (2): Nonetheless, if the new input comes from an out-of-domain distribution which is drastically different from the training data—we can adapt CM to this case with the help of a lightweight pretrained model.
> To support **progressive personalization**, our method can be designed to be **continually updatable**:
> *When a new input \$X\_{\text{new}}\$ is given, which can be significantly different from our in-domain datasets, we can use a pretrained sentence encoder to compute the similarity between the embedding of \$X\_{\text{new}}\$ and those of existing user-preference samples with predefined candidate sets. If the similarity score is lower than a predefined threshold (e.g., 0.1), the system flags \$X\_{\text{new}}\$ as a potential out-of-domain sample.*
> The system then passes the query to the LLM without showing predefined classes. If user feedback is provided either through explicit correction or satisfaction signals the user may reject the undesired response \$Y'\$ and supply a preferred alternative \$Y\$. This new instance–response pair is incorporated into the memory \$M(X)\$ without retraining the LLM. If the user is satisfied, the memory bank records the preference as aligned; otherwise, it logs the misalignment. This setup supports **continuous learning without fine-tuning** and enables enhanced personalization via a lightweight, **per-user memory**. We agree this is a valuable area to explore. **However, we considered it to be outside the immediate scope of this paper.**
>
> ---
>
> #### **W2 The claim that LLM is ineffective i,..., with reasonable compute.**
>
> We acknowledge that the term “ineffective” was too strong and will revise it to better reflect our intended claim: while existing methods may be effective for fine-tuning on a per-user basis, they become infeasible when deployed across thousands of users due to storage overhead, high expenses, and the time-consuming nature of fine-tuning.
>
> #### **W2.1 Parameter-Efficient Fine-tuning (e.g., LoRA)**
>
> Our method leverages a **small number of preference samples** to estimate **instance–response-dependent discrepancies**, **without any model weight updates**. As demonstrated in our experiments, this lightweight memory-based method improves user alignment effectively **without fine-tuning**, offering a scalable alternative for real-world applications.
>
>
> #### **W2.2 In-Context Learning (ICL)**
>
> In-context learning allows users to provide preferences as part of the prompt. While this works in single-user settings, it is **not scalable in practice** due to several limitations:
> * It requires **users to provide their preferences** per query,
> Our approach addresses these limitations by estimating **instance–response discrepancies** from a small set of user-preference samples. These discrepancies are stored in a **personalized memory bank** that:
> * Requires **no model updates**,
> * Is **lightweight and can be updated**,
> * Enables **silent personalization** without needing to repeatedly asking the user.
>
> ---
>
> #### **W2.3 Redundancy on pages 6–7**
> We have **tighten the section on pages 6–7** and **moved extended explanations to the appendix**.
>
> ---
>
> #### **Additional Comments**
> 1. We have revised the introduction and removed "hallucination".
>
> 2. Problem setup has been moved forward; related work moved to the end.
>
> 3. \$Y\$ is defined at the response level.
>
> 4. Notation on page 8 revised.
>
> 5. We acknowledge the idealized assumption of noise-free preferences and note it can be relaxed. We have revised it accordingly in Section 4 on page 8.
>
> 6. Obvious improvement results are now bolded only; (In Table 3 we have shown the exact improvements)
>
> 7. CIs are reported as mean ± std over 3 seeds.
>
> ---
> Please let us know if any further clarifications are needed.

---

### Review · Reviewer_Aei3 · 2025-07-06

**Summary Of Contributions:**

The paper introduces a new method Consistent Marginalization which is a technique for aligning LLM outputs with user-preference. The method constructs a memory bank based on discrepancies between LLM generated responses and user preference. The work demonstrates improved performance across a range of datasets and architectures.

**Audience:**

Yes

**Broader Impact Concerns:**

No broader impact concerns.

**Claims And Evidence:**

Yes

**Requested Changes:**

I would recommend using figure 1 to be more illustrative of the method and to ground it with real-world examples (input, generated response, ground truth, etc). Otherwise the method feels disconnected from the single example given of investment. The bottom portion of the figure in section 4.2 is also difficult to parse with the conditional statements inside of the matrix. Overall my recommendations are to improve the readability of the method section and improve the figures to make them more illustrative of what they are trying to convey. I think a figure which clearly demonstrates the method would really improve the paper. The changes are not critical to acceptance.

**Strengths And Weaknesses:**

- The theory/notation, experiments, and example given in the introduction feel a bit decoupled. The example given in the introduction was that the language model should take into account the risk averse nature of the person when suggesting 80% investment in crypto currency. The method section then immediately defines response classes in the problem setup.  It seems that having a finite, defined set of response classes is a large assumption and more concrete scenarios where this applies should be given as grounding to this problem. The initial motivation is that user preference is not factual so it's a difficult problem, but then it seems like many user-preference scenarios wouldn't have a clearly defined set of response classes. In the investment example given, the output could be continuous from 0-100%, the categories for investment are likely an open set. Along this line it would also be helpful to have a figure which illustrates the experimental setup and some of the sample input, the memory banked, and the generated and ground truth responses from one of the datasets.

- From  anecdotal experience it seems that current LLM’s with long context account for preference in long conversations like the example given. If the user says that they are risk averse in the context, then at least anecdotally I find that LLM’s condition their response on this type of information. It would be good to address this in the motivation of the introduction and potentially in the experiments why information in the context is not sufficient for user preference.

- Overall the paper has thorough experiments and supports their claims with reasonable evidence. The method is interesting and appears to have strong empirical results, though I am not as familiar with the benchmarks and methods in this area. The weakness I would highlight are that the method section is difficult to parse, the method could be better grounded with some qualitative examples, and the problem could be better motivated in the introduction.

---

> ### Author Response · Authors · 2025-07-20
> **Response to Reviewer Aei3**
>
> Dear Reviewer
>
> Really thankful for your time in reviewing our paper and providing us detailed and constructive feedback.
>
> ---
>
> #### **W1.1: The theory/notation, experiments, and example given ,..., 0-100%, the categories for investment are likely an open set.**
>
> We thank the reviewer for pointing out the mismatch between our motivating example and the discrete setup in our method. In response, we have updated the example in the introduction (page 2) to better reflect this. Please refer to the revised introduction for details. In short, we now use a regional slang example to illustrate how our method handles preference alignment. For instance, a user may say "Kopi O" to request black coffee with sugar. This highlights the importance of estimating discrepancies between users with distinct backgrounds and the LLM. Our approach enables effective response alignment without requiring fine-tuning of the underlying LLM.
>
> ---
>
> #### **W1.2: Along this line, it would also be helpful to have a figure illustrating the experimental setup and some of the sample input, the memory banked, and the generated and ground truth responses from one of the datasets.**
>
> Thanks. We have added **Figure 4** on page 7 to the revised version of the manuscript that illustrates the experimental setup. This visualization includes:
> * A sample input query
> * The memory banked (i.e., previously collected discrepancy-aligned responses)
> * The response generated by the LLM
> * The final selection aligned with the user’s preference
> * The ground-truth response from one of the benchmark datasets
>
> ---
>
> #### **W2.1: From anecdotal experience it seems that current LLM’s ,.., that LLM’s condition their response on this type of information.**
>
> We thank the reviewer for this important observation. It is true that some proprietary chatbots (e.g., ChatGPT with memory) with long context windows are able to condition on user preferences. However, relying solely on in-context memory has several **critical limitations** that our method is designed to overcome:
>
> 1. **Lack of persistence across sessions**
>    In **API-based settings** that each query is independent. User context is not preserved between sessions, meaning that preferences must be repeated every time. Our approach enables *persistent memory* across queries without requiring conversational history to be resent or reprocessed.
> 2. **No mechanism for correcting prior misalignment**
>    Even in long-context conversations, the model may produce misaligned responses. Without an external mechanism to log these discrepancies, users must manually intervene. Our method **records and learns from these mistakes**, using them to improve future alignment.
> 3. **Inaccessibility of proprietary memory**
>    While commercial LLMs (e.g., ChatGPT or Claude) may maintain internal memory for logged-in users, this **memory is not transferable or auditable**. In contrast, our approach **decouples personalization from the LLM provider** by storing user preferences in a lightweight, portable memory bank—making it usable across models, providers, and even offline deployments.
>    * **Across online providers:** A discrepancy memory bank estimated with ChatGPT‑3.5 can be directly applied to ChatGPT‑3.5 or ChatGPT‑4o‑mini, **as demonstrated in our experiments.**
>    * **In offline deployments:** A discrepancy memory bank estimated from Llama‑3‑8B‑Instruct can be reused with the same model running locally, without requiring any further fine-tuning.
> 4. **Scalability and control for organizations**
>    For companies deploying LLMs via third-party APIs, controlling personalization is difficult. Our approach allows them to maintain **per-user memory banks**, ensuring consistency in service quality even when switching providers or models.
>
> #### **W2.2: It would be good to address this in the motivation of the introduction and potentially in the experiments why information in the context is not sufficient for user preference**
>
> We have revised the Introduction (page 2) and the Methodology section (page 7 and 8) to improve clarity. Additionally, we have added Figure 4 on page 7, which includes qualitative examples from the Banking77 and MASSIVE datasets to better support the explanation.
>
> ---
>
> #### **W3.1: The weakness I would highlight are that the method section is difficult to parse, the method could be better grounded with some qualitative examples.**
>
> Thank you for the feedback. We have revised the Methodology section (page 7 and 8) and added Figure 4 on page 7, which provides qualitative examples from the Banking77 and MASSIVE datasets.
>
> #### **Requested Changes.**
>
> To address the requested changes, we have included an illustrative Figure 4 on page 7 that showcases examples from the actual datasets used. We have also revised the methodology section for improved readability in the updated manuscript.
>
> ---
> Please let us know if further clarification would be helpful. We sincerely appreciate your valuable feedback.

---

### Decision · Action_Editor_WXzp · 2025-08-30

**Recommendation:** Accept as is

**Additional Comments:**

Despite the scope limitations, this work makes a solid contribution to practical LLM personalization. The discrete response setting, while restrictive, captures many real-world applications (customer service, intent classification, recommendation systems). The experimental validation is thorough, and the authors have adequately addressed methodological concerns raised during review.

The paper represents incremental but valuable progress in personalization techniques, with clear practical benefits for deployment scenarios where fine-tuning is infeasible.

**Audience:**

Yes

**Audience Explanation:**

The initial reviews are a bit mixed, raising concerns about clarity and method grounding, notation concerns, as well as limited applicability. Throughout the rebuttal process, the authors have made substantial revisions to the manuscript, which have addressed most of the concerns from the reviewer.

The remaining concern from Reviewer DHBS is that the method applies specifically to discrete, predefined response sets rather than general LLM applications. While this limits broad applicability, it addresses an important and practical subset of LLM deployment scenarios, and should be of interest to practitioners who work specifically on personalization of LLMs.

**Claims And Evidence:**

Yes

**Claims Explanation:**

This paper proposes Consistent Marginalization (CM), a method for personalizing LLM outputs by maintaining instance-response-dependent discrepancy memory banks. The approach addresses user preference alignment without requiring model fine-tuning, using a lightweight memory-based framework that records misalignments between LLM outputs and user preferences.

The experimental validation demonstrates consistent improvements across most model-dataset combinations, with proper statistical reporting and honest acknowledgment of cases where improvements were limited.